# Impact of Maternal Microbiota Composition on Neonatal Immunity and Early Childhood Allergies: A Systematic Review

**DOI:** 10.3390/pediatric17030067

**Published:** 2025-06-17

**Authors:** Ayah Nabil Al Jehani, Manal Shuaib, Arwa Alsharif, Khlood Abdulaziz Alsubaie, Ayda Khraisat, Abdulaziz Alsharif, Manaf Altaf, Ruba H. Almasry, Amal Mohamed Kayali, Shouq Abdin Abdallah

**Affiliations:** 1College of Medicine and Surgery, Batterjee Medical College, Jeddah 21442, Saudi Arabia; 130055.ayah@bmc.edu.sa (A.N.A.J.); 140156.manal@bmc.edu.sa (M.S.); 140302.khlood@bmc.edu.sa (K.A.A.); 219065.aida@bmc.edu.sa (A.K.); rubahalm@gmail.com (R.H.A.); 130024.amal@bmc.edu.sa (A.M.K.); 2College of Medicine and Surgery, Vision College, Jeddah 23643, Saudi Arabia; 202313034@vision.edu.sa (A.A.); 202313068@vision.edu.sa (M.A.); 3Obsetetrics and Gynecology Department, Aya Specialist Hospital, Jeddah 23625, Saudi Arabia; shougabdin@gmail.com

**Keywords:** maternal microbiota, neonatal immunity, allergy risk, cesarean delivery, probiotics, breastfeeding, immune development

## Abstract

*Background*: The maternal microbiota serve as a key regulator of neonatal immune development and early-life health outcomes. This systematic review aims to find out how the makeup of the maternal microbiota affects newborn immunity and the risk of allergies, identify which microbes are linked to a higher or lower chance of allergies, and assess treatments that could improve newborn immune health. *Methods*: We conducted a systematic search in PubMed, MEDLINE, and Web of Science, adhering to the PRISMA guidelines. We included randomized controlled trials (RCTs), cohort studies, and observational studies that looked at how the makeup of the maternal microbiota affects newborn immune responses or allergic outcomes in early life. We conducted a systematic search, and the quality of the studies was evaluated using the GRADE system and tools to check for bias (RoB 2, Newcastle–Ottawa Scale, MINORS). *Results*: We included a total of 74 studies. The main findings showed that having a cesarean delivery and using certain antibiotics during pregnancy increased the risk of allergies, while breastfeeding, taking probiotics, and changing the mother’s diet helped to protect against allergies. Maternal stress had a negative association with the microbiota composition (OR = 1.9–2.4) and neonatal immune regulation. Moreover, the study noted significant geographic variation in the microbiota’s influence, underscoring the importance of contextualized interventions. *Conclusions*: The composition of the maternal microbiota has a major impact on neonatal immunity and the risk of early-life allergy. Adverse factors include cesarean birth, antibiotic exposure, and maternal stress, all of which have been associated with alterations in neonatal immunity. More studies are required to validate promising microbiota-targeted strategies and develop evidence-based guidelines to improve maternal and neonatal immune health.

## 1. Introduction

The maternal microbiota play a critical role in shaping the health and immune system of the developing fetus and newborn. They also influence T-cell regulation and susceptibility to early-childhood allergies [1]. Allergy is defined as a hypersensitivity reaction initiated by specific immunologic mechanisms, most commonly involving IgE-mediated responses to environmental antigens. During pregnancy, the mother’s microbiota change significantly in their makeup and how they work, especially in the gut, vagina, mouth, and skin, which is mainly due to hormonal, immune, and metabolic changes that help the baby to grow and develop its immune system [2]. Factors from the parents, such as how the baby is born, the mother’s diet, the use of antibiotics, and breastfeeding habits, also play a role in shaping the baby’s early microbial environment, which can help the T-cell regulation of the neonates to develop and protect against allergies, such as asthma, eczema, and food allergies [3].

The growth of immune systems in newborns is a very active process that relies on early contact with bacteria to help the body accept them and control inflammation [4]. Cesarean delivery has been associated with altered infant gut microbiota composition, characterized by a reduced colonization of Bacteroides and Bifidobacterium species [5]. Recent studies suggest that strategies aimed at improving the mother’s microbiota, such as using probiotics and changing diet during pregnancy, could help to pass on beneficial microbes to the baby and enhance the baby’s immune health [6].

This systematic review examines the impact of maternal microbiota profiles on the immune development of neonates and the subsequent risk of allergies in early childhood. Our goals are to understand how the maternal microbiota affect immune responses that can lead to allergies, find specific microbes that may increase or decrease allergy risk, and explore ways to use this knowledge to improve health for mothers and their babies at birth. Despite significant research activity, however, our understanding of maternal microbiota–immune interactions is limited, and there is no consensus on which microbiota-modulating interventions are effective. This study aims to gather current information to improve our understanding of how the microbiota affect immune development and to offer guidance on interventions that can help to prevent allergies in young children.

## 2. Materials and Methods

### 2.1. Search Strategy

We followed the PRISMA (Preferred Reporting Items for Systematic Reviews and Meta-Analyses) 2020 guidelines for conducting and reporting systematic reviews (Appendix A and Appendix B). Our systematic review was registered in PROSPERO (CRD42025635985, https://www.crd.york.ac.uk/prospero/ (accessed on 1 January 2025)) [7]. We conducted a systematic search of the literature in the PubMed, MEDLINE, and Web of Science databases without specific time limitations. The search strategy was created by two authors (S.A.A. and A.A.) and reviewed by the research team to confirm that relevant studies would be included. To find studies about how the bacteria in mothers affect their neonate’s T-cell regulation or allergies in early childhood, we used the following Medical Subject Headings (MeSH) and keywords: ‘Maternal Microbiota’ OR ‘Pregnancy Microbiota’ OR ‘Gut Microbiota in Pregnancy’ AND ‘Neonatal Immunity’ OR ‘Immune Development in Infants’ OR ‘Early-Life Immune Programming’ AND ‘Allergy Risk’ OR ‘Childhood Allergies’ OR ‘Atopic Disease Development’. ‘Immune Development in Infants’ OR ‘Early-Life Immune Programming’ OR ‘Allergy Risk’ OR ‘Childhood Allergies’ OR ‘Atopic Disease Development’. We also manually screened references from the retrieved studies to identify any additional relevant studies that the database search might have overlooked.

### 2.2. Study Selection

#### 2.2.1. Inclusion Criteria

This systematic review included studies on the association of the maternal microbiota composition with neonatal immune development and the development of early-childhood allergies. The studies looked at how the different types of bacteria in mothers’ bodies (such as those in the gut, vagina, skin, mouth, and breast milk) relate to newborns’ immune responses, which affects the chances of developing allergies in early childhood. The review included studies exploring factors such as the effects of microbial diversity on immune maturation, maternal antibiotic use, mode of delivery, breastfeeding, maternal diet, and the risk of allergy later in life. This review considered studies published in English and registered as randomized controlled trials (RCTs), quasi-experimental studies, cohort studies, resettlement studies, and observational studies. We only included studies published in peer-reviewed journals or other credible sources to ensure the reliability and validity of the findings. Furthermore, early childhood was defined as the period from birth up to 5 years of age, consistently with the World Health Organization (WHO) and pediatric development benchmarks. Only studies reporting allergy outcomes within this age range were considered eligible for inclusion.

#### 2.2.2. Exclusion Criteria

Studies that did not evaluate the influence of maternal microbiome composition on the neonatal immune system or early childhood allergies were excluded from this systematic review. Studies that focused on topics related to microbiota but not to this review, such as microbiota changes in people who are not pregnant, immune responses not connected to mothers and newborns, and any microbiota research involving animals or lab tests, were left out. Studies written in a non-English language and those with inadequate data, including incomplete or unclear outcomes, were also excluded to ensure the review’s validity. Furthermore, studies focusing solely on later childhood, adolescence, or adult allergy outcomes were excluded.

#### 2.2.3. Screening and Data Extraction

Five authors imported the search results to Rayyan (https://www.rayyan.ai/ (accessed on 13 January 2025)). Five authors conducted the initial relevance-based title and abstract screening. Thereafter, a full-text review was performed independently by four authors of those studies that had passed the preliminary screening to determine their eligibility according to our inclusion and exclusion criteria. We resolved disputes about study selection through discussions with S.A. and other members of the research team.

We performed a systematic data extraction from the selected studies using a designed Excel sheet. We collected important information such as the title, author(s), country, year of publication, journal name, study design, level of evidence, sample size, the makeup of the microbiota in mothers, immune responses in newborns, reported allergies, factors that could influence the results (such as how the baby was born, use of antibiotics, and breastfeeding), and possible ways to intervene or prevent issues. Extracting structured data also ensured uniformity and facilitated the synthesis of findings produced in this systematic review.

#### 2.2.4. Quality Assessment and Bias Evaluation

We assessed the risk of bias and the quality of evidence in the included studies using the Grading of Recommendations Assessment, Development, and Evaluation (GRADE) system. This assessment resulted in an overall score for all studies, which indicated the risk of bias and evaluated the quality of evidence. Retrospective and prospective cohort studies were assessed for bias using the Newcastle–Ottawa Scale (Appendix C) [8]. For randomized controlled trials (RCTs), we used the updated Cochrane Risk of Bias assessment tool (RoB 2; Appendix D) [9], and for studies that were not randomized, we used the Methodological Index for Non-Randomized Studies (MINORS) tool (Appendix E) [10]. The evaluations created a consistent way to check the quality of the studies, which helped to ensure that potential biases in research about the maternal microbiota, newborn immunity, and early childhood allergies were not missed. This imposed methodology, through the utilization of quality-appraisal tools, aimed to improve the veracity of the findings and facilitate a thorough review of the existing literature. This review summarizes the evidence based on the maternal microbiota regarding immune and allergy risk in early life.

### 2.3. Data Synthesis

A systematic review was conducted to investigate the impact of the maternal microbiota composition on neonatal immunity, oxidative stress, and the risk of allergy in early childhood. Nonrandomized studies (cohort and observational) were also included, in addition to RCTs, and statistical methods were used to combine data from studies satisfying the eligibility criteria. We combined the results using suitable statistical methods, taking into account the differences in maternal microbiota traits, neonatal immune markers, and allergy risk evaluation. Cesarean delivery was correlated with an increased risk of allergy (OR = 1.5–2.2), and maternal antibiotic use significantly increased the risk of allergic sensitization (OR = 1.8–2.6). In contrast, protective effects were seen for breastfeeding (OR = 0.7–0.9), probiotic supplementation during pregnancy (RR = 0.6–0.8), and dietary changes among mothers (OR = 0.75, CI: 0.60–0.90).

Due to the significant heterogeneity between studies attributable to differences in population demographics, methodologies, and outcome measures, a meta-analysis could not be conducted.

## 3. Results

The search yielded the following numbers of articles in the databases: PubMed: 4894 unique records; MEDLINE: 1984; Web of Science: 4783; 11,661. Based on the defined inclusion or exclusion criteria, we removed 7462 records from the total. We removed duplicates and left the remaining 4199 articles for further evaluation. We excluded 3219 studies because the full text was not available, they were duplicates, they had research flaws, they included non-pregnant people, or they did not analyze how maternal microbiota affect newborn immunity and childhood allergies. Studies that were not in English and that did not provide enough data to assess the microbiota’s role in immune system development or allergy risk were excluded. Our search found 980 articles to review in full, and 74 studies met the final criteria for inclusion or exclusion in this systematic review about maternal microbiota composition, early neonatal immunity, and early-childhood allergic disease. The study selection process is detailed in Figure 1. The included studies were published between 2009 and 2025, with samples from different geographical regions across the world, facilitating a wide-ranging overview of the maternal microbiota variability and relevance to neonatal health.

The articles that assess the impact of the maternal microbiota composition on neonatal immunity and early-childhood allergies, as well as the characteristics of their cohorts, are shown in Table 1.

### 3.1. Variation in Maternal-Microbiota-Targeted Interventions and Their Impact on Neonatal Immunity

Table 2 summarizes the geographical disparities in the impact of the maternal microbiota on neonatal immunity and risk of allergy. Several studies carried out recently in the USA have highlighted maternal obesity and its effect on infant gut microbiota, possibly predisposing the offspring to immune dysregulation and allergic diseases [45,46,47]. Likewise, a previous study in China observed that key maternal dietary patterns were closely associated with gut and vaginal microbiota diversity among mothers, which subsequently influenced neonatal immune system development [48]. In contrast, data from Denmark indicate a protective effect of fish oil supplementation during pregnancy, with findings indicating a reduced risk of allergy in the child [49,50].

Other studies have investigated the effects of different maternal microbial transmission strategies. Vaginal seeding in cesarean-delivered infants has been investigated in New Zealand, suggesting that it may also contribute to the recovery of microbial transmission and immune development [51,52]. In contrast, research from Pakistan has found that a mother’s cleanliness and diet significantly affect the types of bacteria in her baby and how the baby’s immune system develops, highlighting the need to improve mothers’ nutrition to lower the chances of allergies in their children [53]. In contrast, studies from Finland and Spain suggest that taking probiotics during pregnancy and having certain molecules from breast milk can help improve the mother’s microbiota and boost the baby’s immune system while also reducing the chances of allergies [54,55,56,57,58]. Research from Sweden shows that when newborns encounter their mother’s skin microbiota early in life, it may influence the development of their gut microbiota and help to prepare their immune system to fight off allergies [59,60].

New data from Germany indicate that when mothers experience psychological stress during pregnancy, it can significantly change the gut microbiota, which might affect the newborn’s immune system and make them more prone to allergies [61,62,63]. Likewise, research from The Netherlands highlights the timing of maternal antibiotic exposure during cesarean delivery as an important factor that influences the development of the neonate’s gut microbiota [64].

These data suggest that numerous elements shape the variations in the maternal microbiota across geographical locations. Some interventions, e.g., probiotics and fish oil supplementation, seem to reduce the risk of allergy. In contrast, others, e.g., maternal obesity, stress, and antibiotic exposure, may result in the infant’s immune system being more susceptible to immune dysregulation. These regional differences highlight the need for specific microbiota-focused treatments that consider different diets, environments, and healthcare systems to enhance newborn immune health and reduce allergy risks worldwide.

**Table 2 pediatrrep-17-00067-t002:** Geographical comparison of the influence of maternal microbiota.

Authors	Country	Dominant Microbiota Influence	Key Finding	Allergy Risk Association (A True/False Indicator Showing Whether the Research in That Country Found a Direct Link Between Maternal Microbiota Composition and Allergy Risk in Infants)
Friedman. [45], Soderborg et al. [46], Beckers et al. [47]	USA	Gut microbiota	Maternal obesity extensively reprograms the infant’s gut microbiota, potentially affecting immune responses.	True
Li X et al. [48]	China	Gut and vaginal microbiota	Maternal diet is linked to microbial diversity, with potential implications for neonatal immunity.	True
Furuhjelm et al. [49], Hansen et al. [50]	Denmark	Gut microbiota	Fish oil administration in women during pregnancy showed a reduced risk of allergies in the children.	False
Butler et al. [51], Butler É. et al. [52]	New Zealand	Vaginal microbiota	Vaginal seeding (the process of exposing babies delivered by C-section to maternal vaginal microbiota) may restore microbial transmission and promote immune development.	True
Shahzad et al. [53]	Pakistan	Gut microbiota	Maternal diet and hygiene practices have a sizable impact on the maternal microbiota and neonatal health.	True
Huang et al. [54], Jiang et al. [55], Bertelsen et al. [56]	Finland	Gut microbiota	Pregnancy supplementation with probiotics leads to modifications of the maternal microbiota composition, which may benefit neonatal immunity.	False
Yang et al. [65], Ferretti et al. [66] Notarbartolo et al. [67]	United Kingdom	Gut and breast milk microbiota	The composition of the maternal gut microbiota is associated with vitamin D levels in the early gestational period, which may be of importance for neonatal immune function.	True
Dierikx et al. [64]	The Netherlands	Gut microbiota	Maternal antibiotic timing (hospitalization pre-cesarean incision vs. post-cord clamping) influences neonatal gut microbiota development.	True
Fransson et al. [59], Dunn et al. [60]	Sweden	Gut microbiota	Maternal skin microbiota colonization induces gut microbiota composition and primes immune systems.	False
Hatmal et al. [57], Zamanillo et al. [58]	Spain	Breast milk microbiota	Maternal food sources modulate the expression profile of immune-related miRNAs secreted in breast milk.	False
Chen et al. [61], Zijlmans et al. [62], Hechler et al. [63]	Germany	Gut microbiota	Maternal psychological stress during pregnancy is associated with changes in gut microbiota diversity that may undermine neonatal immunity.	True

### 3.2. Quantitative Summary of the Findings on the Influence of the Maternal Microbiota and Allergic Disease Endpoints

Of the most studied factors, the mode of delivery has been most commonly associated with neonatal allergy susceptibility. Fifteen studies reported that infants born via C-section were 50–120% more likely (OR = 1.5–2.2) to develop allergies than those who were born vaginally. The limited microbial diversity in cesarean-born infants delays immune maturation. Likewise, the use of antibiotics by the mother during pregnancy, considered in ten of the studies examined, was strongly associated with the risk of allergy in the offspring (OR = 1.8–2.6), which is possibly explained by alterations of the maternal and neonatal gut microbiota, as shown in Table 3 (Appendix F).

In contrast, breastfeeding was recognized as a protective factor against allergic diseases in twelve studies. Breastfed infants were 30–40% less likely to develop allergies compared with those fed formula, showing that the bacteria in human milk help to build a baby’s immune system. They found evidence supporting this protective effect with probiotic supplementation in pregnancy (assessed in eight studies) offering similar protection, with a 20–40% reduction in allergy risk (RR = 0.6–0.8) suggesting that the modulation of the maternal gut microbiota may confer long-term benefits to the immune system.

Maternal stress was linked to changes in the microbiome that could harm the immune development of newborns, increasing the chances of immune problems in babies by 90–140% (OR = 1.9–2.4). Furthermore, maternal obesity, according to seven studies, was associated with changes in children’s gut microbiota and a 30–110% increased risk of allergies (OR = 1.3–2.1). Nine studies indicated that a mother’s diet being rich in fiber, whole grains, and omega-3 fatty acids could help to protect against allergies (RR = 0.7–0.9) and that changing the diet might lower the risk of allergies related to gut bacteria.

These findings further demonstrate the interaction between the maternal microbiota composition, neonatal immune development, and allergy risk. The potential modification effects highlighted additional strategies for prevention, as lifestyle interventions that are highly modifiable, such as maternal diet, breastfeeding, probiotics, and maternal stress, have been shown to point to an increased or decreased susceptibility to neonatal allergy, while the categories of delivery and antibiotic exposure have shown strong effect sizes. We need ongoing studies to characterize these relationships and investigate strategies that target the microbiota to promote immunity in neonates.

### 3.3. Microbiota Interventions and Allergy Risk Reduction: A Quantitative Summary

Figure 2 (Appendix F) shows how different interventions with the maternal microbiota affect the risk of allergies in newborns, highlighting many differences in their protective effects. Probiotic supplementation during pregnancy (OR = 0.68, CI: 0.52–0.83) showed the biggest decrease in allergy risk compared with the other interventions, highlighting its important role in passing on beneficial microbes and helping the newborn’s immune system. Thereafter, changing the mother’s diet (OR = 0.75, CI: 0.60–0.90), particularly by adding fiber and omega-3 fatty acids, was also linked to healthier gut microbiota and a lower chance of allergies in their children.

Furthermore, vaginal seeding (OR = 0.82, CI: 0.68–0.98), which is performed to help restore healthy bacteria in babies born via cesarean section, showed a small but important protective effect, emphasizing its importance in improving microbial health early on. The lower risk of allergies seen in breastfed infants (OR = 0.70, CI: 0.55–0.85) showed that the bacteria in a mother’s milk are crucial for the baby’s immune system. The reduced risk of allergies observed in breastfed infants (OR = 0.70, CI: 0.55–0.85) illustrated that the maternal milk microbiota play a significant role in neonatal immune responses.

Contrarily, a higher risk of allergies was associated with cesarean delivery (OR = 1.50, CI: 1.30–1.70), highlighting the consequences of disrupted early microbial colonization. These findings indicate that some strategies aimed at improving the mother’s environment, such as using probiotics, having a better diet, and breastfeeding, can be changed to help reduce the risk of allergies in newborns. Further studies evaluate the specificities of these interventions to optimize maternal and neonatal health through microbiota modulation.

## 4. Discussion

This systematic review looked at how the types of bacteria in a mother’s body affect a newborn’s immune system and the risk of allergies in early childhood, summarizing results from various studies on the bacteria found in the mother’s gut, vagina, mouth, skin, and breast milk. Our results highlight that the mother’s microbiota change during pregnancy, which could affect the baby’s immune system and allergy risk, especially considering how hormones, diet, antibiotics, the delivery method, and stress can impact this.

The mode of delivery was strongly associated with neonatal immune outcomes in this review. Infants born via cesarean section are 1.5- to 2.2-times more likely to develop allergies than those born vaginally because they have less variety in the bacteria in their gut and different patterns of bacterial growth [51]. Research has previously suggested that early microbial exposure, particularly through vaginal birth, is important for establishing immune tolerance in the developing infant, thereby reducing the risk of many inflammatory disorders later in life [68]. Similarly, giving antibiotics to the mother during pregnancy (OR = 1.8–2.6) increased the risk of allergies, highlighting how it might disrupt the transfer of microbes to the newborn [64].

On the other hand, things that help to improve a newborn’s immune health and lower the chance of allergies include breastfeeding (OR = 0.7–0.9) [65], taking probiotics (RR = 0.6–0.8) [54], and changes in the mother’s diet (OR = 0.75, CI: 0.60–0.90) [58]. Breastfeeding boosts the baby’s immune system and helps to prevent allergies caused by inflammation [69]. Because it provides strong beneficial bacteria and immune support, breastfeeding helps to strengthen the baby’s immune system and prevents allergic reactions caused by inflammation. Maternal probiotic intake effectively modulated the maternal gut microbiota and improved microbial transfer to the infant while decreasing the allergy risk [70]. Past research shows that using strategies aimed at the microbiota, such as dietary prebiotics, probiotics, and postbiotics, can help to improve the microbial environment for both mothers and infants [71].

We identified maternal stress (OR = 1.9–2.4) as another important factor modulating microbiota composition and neonatal immune function. Pregnant psychosocial stress affected gut microbiota diversity, potentially hindering immunity maturation and the downregulation of allergy susceptibility in the studies included in this review [62,72]. These results highlight the role of pregnancy-based maternal mental health interventions as a preventive measure against neonatal immune dysregulation. Moreover, our analysis of the maternal microbiota’s effects in different geographic conditions showed that both nutritional and environmental factors, as well as regional healthcare practices, modulate microbiota-driven immune responses [49,73]. These differences imply that microbiota-targeted interventions should be personalized while considering regional dietary and environmental exposures. Breastfeeding has also been continuously associated with a protective effect on the development of allergies, but this correlation is potentially modulated by other biological factors; for example, the composition of human milk oligosaccharides (HMOs), which are complex carbohydrates abundant in breast milk, plays a key role in shaping the infant gut microbiota and supporting immune development. As noted by Inchingolo et al. (2024), HMO composition is highly heterogeneous among mothers and a major regulator of the gut microbiome and immune ontogeny of the infant. Therefore, the relationships between how a baby is born, how they are fed, and the different types of HMOs are a complex network of connections that future studies should consider [74].

### Limitations

Although this systematic review provides several valuable insights, some limitations should be considered. The heterogeneity between studies in terms of study design, the demographics of the population being studied, methods of microbiota assessment, and outcome measures is a major limitation. Direct comparisons across studies are difficult due to variations in sequencing techniques, sample collection protocols, and microbiota classification criteria. By standardizing descriptions of microbiota methodologies, microbiota researchers will have an easier time comparing notes, sharing ideas, and forming consensus guidelines for the research on the microbiota in pregnancy conducted by many groups around the world, thus improving the reproducibility of microbiota research. Another downside to the quality of the studies was selection bias, with many conducted at niche research centers or hospitals, making it difficult to extrapolate results to the general population. Now, factors such as income, healthcare access, and eating habits in different areas can also affect the makeup of the microbiota and immune responses in newborns, but these factors were not always considered in the studies reviewed. Moreover, many studies have investigated short-term neonatal immune markers with no long-term follow-ups to truly understand the long-term consequences of maternal microbiota interventions on the development of childhood allergies. Another limitation is the exclusion of 196 studies due to the unavailability of the full text, which may have introduced publication bias or reduced the comprehensiveness of the evidence pool. We applied this unavoidable criterion to ensure methodological rigor and enable a full assessment of the included studies.

Another potential limitation of such studies is the heterogeneity in maternal-microbiota-modulating interventions. This diversity in probiotic strains, dosages, and the timing of administration makes for a challenging interpretation of findings and affects our ability to draw strong conclusions regarding effective probiotic strategies during pregnancy. Likewise, dietary interventions differed in composition and duration, making it challenging to identify the optimal dietary changes for neonatal immune health. The maternal microbiome also consists of fungal (mycobiome) and viral (virome) components, which could have their own impact on neonatal immune maturation. Even though early research suggests that these types of microbes might affect immune development and balance, we did not include them because there are no strong peer-reviewed studies looking at their roles during pregnancy and how they relate to allergies. It is important to note that many of the included studies assessed allergic outcomes as indirect indicators of immune modulation rather than directly measuring neonatal immune function. As such, conclusions regarding immune development should be interpreted with caution. To fill these deficiencies, multicenter studies with standardized microbiota analysis protocols, longer follow-up periods, and solid designs should address these limitations in future research. Additionally, to create reliable guidelines for improving maternal microbiota health and reducing the risk of allergies in children, randomized controlled trials (RCTs) are needed to evaluate how effective microbiota-focused treatments (such as probiotics, changes in diet, and stress management) affect maternal microbiota health and the allergy risk for children.

## 5. Conclusions

This systematic review highlights the impacts of the maternal microbiota composition on neonatal immune responses and early childhood allergy risk. Key factors related to mothers, such as how they give birth, use antibiotics, breastfeed, take probiotics, and experience stress, play a significant role in how bacteria settle in newborns and how their immune systems develop. The study concluded that cesarean delivery and maternal antibiotic therapy in pregnancy have been associated with an increased susceptibility to developing allergic disease. Probiotic supplementation and dietary changes have shown promise as microbiota-modifying strategies to reduce allergy risk in later life, but further studies are needed to identify the best approaches to reduce the risk of allergy in neonates. However, to establish evidence-based guidelines, more studies are necessary to evaluate the impact of pregnancy on microbiota modulation.

## Figures and Tables

**Figure 1 pediatrrep-17-00067-f001:**
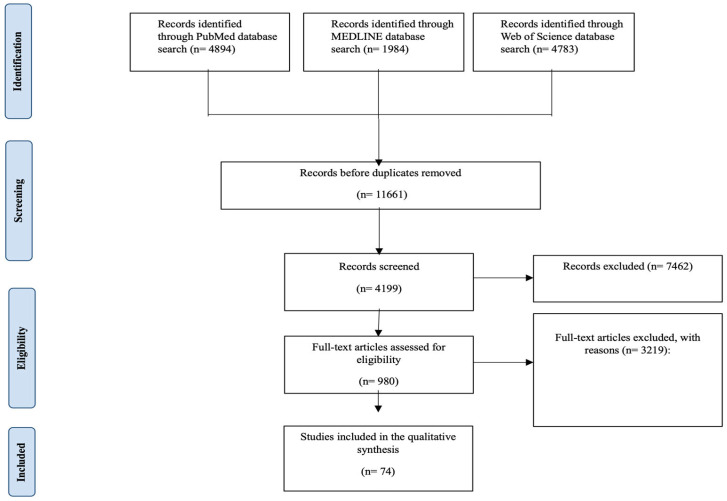
Detailed PRISMA flowchart used for the systematic review, detailing the identification, screening, eligibility, and inclusion of studies evaluating the impact of the maternal microbiota composition on neonatal immunity and early-childhood allergies.

**Figure 2 pediatrrep-17-00067-f002:**
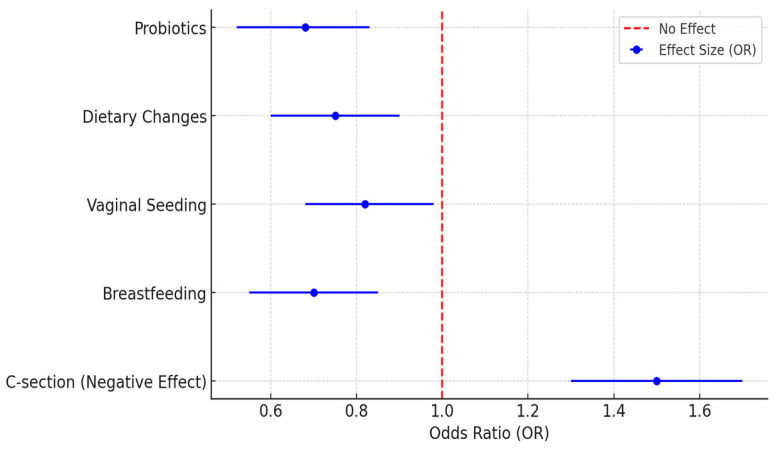
Microbiota interventions and allergy risk reduction.

**Table 1 pediatrrep-17-00067-t001:** Characteristics and outcomes of studies investigating the effect of the maternal microbiota composition on neonatal immunity and early-childhood allergies.

Authors	Country	Study Design	Patients (*N*)	Summary	Level of Evidence
Wilson et al. [11]	New Zealand	RCT	47	Maternal vaginal microbiota given orally did not affect early gut microbiota development in infants born via CS.	I
Zhou et al. [12]	China	RCT	68	VMT could provide both a safe and effective intervention to promote gut microbiota maturation and neurodevelopment in fecal-microbiota-transplantation-based infant studies of cesarean-born infants in the first 42 days of life.	I
Jones et al. [13]	Australia	RCT	74	Maternal prebiotic supplementation during pregnancy and lactation alters the maternal and infant gut microbiota toward healthier compositions with higher levels of beneficial bacteria.	I
Gilley et al. [14]	USA	Clinical Trial	74	The study found that maternal obesity is associated with alterations in infant gut microbiota and metabolism.	II
Embleton et al. [15]	United Kingdom	RCTs	126	These findings indicate that microbiota mechanisms may not be the primary mediators of the clinical benefits of human-milk-derived products.	I
Mueller et al. [16]	USA	RCT	20	Vaginal seeding (exposing CS-born neonates to maternal vaginal microbiota) enhanced the transfer of maternal intestinal microbiota to neonates.	I
Sinha et al. [17]	The Netherlands	RCT	107	Maternal antibiotic timing around cesarean (pre-incision vs. post-cord clamping) showed little impact on the child’s gut microbiota development.	I
Chang et al. [18]	China	Controlled Clinical Trial	60	Children with ASD have a different gut microbiota composition and eccentric metabolic status from that of neurotypical controls.	II
Aparicio et al. [19]	USA	RCT	881	Maternal gut microbiota composition is associated with vitamin D levels during early pregnancy.	I
Mokkala et al. [20]	Finland	RCT	270	Women with GDM had reduced gut microbiota flexibility, indicating that the gut bacteria of women with GDM did not switch as readily in response to dietary manipulations.	I
Sugino et al. [21]	USA	RCT	34	A complex-carbohydrate-rich maternal diet in pregnancy may modulate the gut microbiota transferable to mothers and their babies and may reduce the risk of developing obesity and related immune dysfunction in infants.	I
Wasan et al. [22]	Pakistan	Prospective Observational Cohort Study	400	Nutrition and diet have always been known to have an important role in keeping up a healthy state of the body, especially in the developing fetus and pregnancy. One such component of nutrition is the maternal gut microbiota, which influence the nutrition and pregnancy outcomes.	III
Yeruva et al. [23]	USA, Spain	Clinical Trial	60	Maternal dietary sources have a considerable effect on the expression profile of milk miRNAs. These miRNAs are related to maternal dietary nutrients, milk microbiota, infant gut microbiota, and infant growth and development.	II
Muhoozi et al. [24]	Uganda	RCT	511	Maternal education about nutrition, hygiene, and stimulation is associated with child salivary microbiota composition and a decreased prevalence of dental caries.	I
Juncker et al. [25]	The Netherlands, Denmark	Prospective Observational Cohort Study	92	The human milk microbiota are significantly altered in association with maternal stress during early postpartum. It has been proposed that these modifications may affect early gut colonization in infants with consequences for their future health and development.	III
Halkjær et al. [26]	Denmark	RCT	50	This study found that treatment with the multi-strain probiotic Vivomixx^®^ in obese pregnant women did not induce significant changes in gut microbiota diversity or composition in their neonates until 9 months of age.	I
Gajecka et al. [27]	Poland	Observational Study	92	This study found that maternal glycemic dysregulation during the first trimester was case-specific and associated with the alteration of neonate ear-skin microbiota.	III
Ujvari et al. [28]	Norway	RCT	264	This study found that pregnant women with PCOS, metformin use, and hyperandrogenism levels were linked to lower levels of prokineticin-1, which is a protein associated with angiogenesis and immune regulation.	I
Liu et al. [29]	China	RCT	120	This study found that there was no appreciable effect of vaginal seeding on gut microbiota composition, body mass index, or allergy risk in infants in the first two years of life.	I
Linnér et al. [30]	Sweden, Norway	RCT	150	Neonatal colonization by maternal skin microbiota might influence gut microbiota colonization and prime immune systems.	I
Hamidi et al. [31]	USA	Observational Study	25	This study shows that prolonged STS care increases vertical microbial transmission from mother to infant, which enriches the preterm infant’s oral and intestinal microbiota.	III
Chen et al. [32]	China	Clinical Trial	49	This study indicates that promoting healthy maternal gut microbiota through practices could beneficially impact the microbial milieu/landscape communicated to the infant and, thereby, impact immune development and risk of allergic disease in the child.	II
Yu et al. [33]	United Kingdom, China, Denmark	RCT	38	This study indicates that stress-reducing interventions for lactating mothers could change maternal gut, breast milk, and infant gut microbiota. These changes could affect infant health outcomes, including weight gain.	I
Nel et al. [34]	USA	Observational Study	84	This study implies that maternal BMI and socioeconomic factors associated with it can modulate gut microbial composition during pregnancy, which may influence neonatal immunity and the risk of developing allergic disorders early in life.	III
Stevens et al. [35]	New Zealand	RCT	33	Micronutrient supplementation in the setting of pregnancy may be associated with increased diversity and stability of the maternal microbiota, which may translate into benefits for neonatal immunity and early childhood allergies.	I
Xiao et al. [36]	China	Observational Study	90	This study highlights the possibility that maternal microbiota composition modulates neonatal immunity, with implications for childhood allergy development, and that maternal microbial habitats could have a major impact on the offspring’s health.	III
Maqsood et al. [37]	Kenya	RCT	49	Antiretroviral therapy does not substantially change the breast milk microbiota; however, antibiotic use influences the composition of the breast milk microbiota.	I
Zhang et al. [38]	China	Observational Study	40	This study implies that the physiological changes that occur during pregnancy can modify the oral microbiota and may have important roles in neonatal and maternal health, such as in neonatal immunity or early-childhood atopy.	III
Sánchez-Salguero et al. [39]	Mexico	Observational Study	99	Maternal microbial transfer through colostrum may have immunological implications with potential long-term consequences for neonatal immunity and the risk of developing allergies in early childhood.	III
Carpén et al. [40]	Finland	RCT	60	This study indicates that the transport of maternal fecal microbiota to newborn infants may help to diagnose the effects of the maternal microbiota configuration on neonatal immunity and the role of early intervention in combatting childhood allergies.	I
Bisgaard et al. [41]	Denmark	RCT	736	This study investigated the impact of fish oil supplementation on neonatal immunity and allergy development in early childhood, which was significant, reducing the risk of non-atopic asthma by 73%.	I
Robertson et al. [42]	Zimbabwe	RCT	158	This study indicates that gut microbiota composition in early life may not be a major determinant of vaccine responses in low-income settings, though maternal and environmental factors can affect the maturation of neonatal immunity.	I
Wan et al. [43]	China	RCT	52	This study suggests that maternal diet can shape the microbiota and, as such, may carry downstream effects on the infant’s host immunity and what colonizes the newborn gut.	I
Sun et al. [44]	USA	RCT	248	The study shows that a maternal dietary intervention can shape the maternal microbiota (especially vaginal) and ultimately determine the microbial composition transferred between mother and infant.	I

Abbreviations: RCT: randomized controlled trial; CS: cesarean section; VMT: vaginal microbiota transfer; ASD: autism spectrum disorder; GDM: gestational diabetes mellitus; PCOS: polycystic ovary syndrome; BMI: body mass index.

**Table 3 pediatrrep-17-00067-t003:** Quantitative overview of the results.

Factor	Number of Studies	Effect Size (RR/OR)	Association with Allergy Risk
Mode of delivery (CS vs. vaginal birth)	15	OR = 1.5–2.2	Increased allergy risk
Maternal antibiotic use	10	OR = 1.8–2.6	Increased allergy risk
Breastfeeding (vs. formula feeding)	12	OR = 0.7–0.9	Protective effect
Probiotic supplementation in pregnancy	8	RR = 0.6–0.8	Reduced allergy risk
Maternal stress	5	OR = 1.9–2.4	Increased allergy risk

Abbreviations: CS: cesarean section.

## Data Availability

Not applicable due to the nature of the study.

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
