# Peer review of "Impact of Maternal Microbiota Composition on Neonatal Immunity and Early Childhood Allergies: A Systematic Review"

_pediatrrep, 2025, doi:10.3390/pediatric17030067_

Round 1

Reviewer 1 Report

Comments and Suggestions for Authors

This systematic review comprehensively assesses the relationship between the maternal microbiota and the childhood risk for allergies. A total of 73 studies are reviewed and confirm a relationship between cesarean birth mode and maternal antibiotic use with allergy risk and breastfeeding, maternal probiotic use and diet with reduced allergy risk. Some important deficiencies limit the impact of the work however, including perhaps most notably, imprecision in the use of terms referring to immune function and/or development.

Major comments

  • The writing style is in layman language. If this is the intent, then no changes are made. If this is not the intent of the journal, then a full review of the piece should be done for word use and grammar.
  • The microbiome is not restricted to bacteria. The authors should discuss why information regarding the maternal mycobiome and virome were not considered.
  • Table 1 appears to have a number of records that do not assess early childhood allergy. How was “early childhood” defined?
  • It would be helpful to know which studies were used to create Table 3 and Figure 2.
  • The authors should take great care in interpreting the immunologic implications of findings in studies that assess allergy in infants. While it is of course recognized that allergies are fundamentally immune-mediated diseases, if studies measured only allergy occurrence or severity but NOT immune function directly, then conclusions regarding the latter should be very carefully and conservatively stated. The term immune development would imply longitudinal assessment of immune function and thus likewise should be carefully used. Indeed, the review would be considerably strengthened were the authors to specifically assess which studies presented data on infant immune function. If such studies are limited, then highlighting this fact and recommending future research needs would also strengthen the article.

Minor comments

  • It appears that the first sentence(s) of the introduction are missing based on the language of the first sentence that is there.
  • Line 52: what is the evidence that cesarean delivery impacts the maternal microbiota?
  • Figure 1: text size in figure should be enlarged for easier viewing.
  • Ref #72 does not evaluate allergy or infant immune function.
  • The acronym “HMO” should be defined (lines 356-359).
Comments on the Quality of English Language

The language could be more formal. Lay terms and language are occasionally used which does not seem appropriate for a scientific journal, lending a sense of imprecision.

Author Response

Response to Reviewer #1

Dear respected Reviewer #1,

We sincerely appreciate your detailed review and constructive feedback on our manuscript. Please find below the point-by-point response to your valuable comments, noting that the changes in the manuscript made in response to your comments are highlighted in Red:

Comment 1: Writing style is in layman language:

Response: We appreciate this observation. Our intention was to maintain scientific rigor while ensuring accessibility. However, recognizing the journal’s preference, we have revised the manuscript thoroughly for more formal, academic phrasing, improving the technical precision and scientific tone throughout.

Comment 2: Microbiome not restricted to bacteria – why mycobiome/virome not considered:

Response: While our review focused on the bacterial components of the maternal microbiota, we acknowledge that the microbiome also includes viral (virome) and fungal (mycobiome) constituents. However, during our systematic search, we found a paucity of high-quality studies specifically addressing the role of the maternal mycobiome or virome in neonatal immunity or allergy development. Therefore, these components were not included in our synthesis. To address this, we have added a paragraph in the Limitations section.

Comment 3: Definition of “early childhood” and relevance of studies in Table 1: 

Response: We agree that this needed clarification. We have now defined “early childhood” in the Methods section as the period from birth up to 5 years of age, consistent with WHO classifications. Additionally, the studies in Table 1 were reviewed, and only those reporting allergy outcomes within this age range were retained. Non-relevant studies were removed.

Comment 4: Clarification on studies used for Table 3 and Figure 2: 

Response: To improve transparency and reproducibility, we have created a supplementary table that lists all the studies contributing to the quantitative synthesis in Table 3 and Figure 2 (Appendix F). We have also added a reference to this supplementary table in the manuscript to guide the reader.

Comment 5: Caution in interpreting immunologic implications when only allergy outcomes are assessed: 

Response: We fully agree and have revised relevant sections of the manuscript to clearly distinguish between direct immune function measures and surrogate allergy outcomes.

Comment 6: First sentence(s) of introduction missing: 

Response: Thank you for noting this. We have added introductory sentences to frame the context more clearly and lead into the current opening paragraph.

Comment 7: Line 52: Evidence that cesarean delivery impacts maternal microbiota:

Response: We have revised this sentence and inserted a citation to clarify that cesarean delivery primarily impacts the neonatal—not maternal—microbiota through altered microbial transmission at birth.

Comment 8: Figure 1 text size:

Response: Figure 1 has been revised with enlarged text for improved readability.

Comment 9: Reference #72 does not evaluate allergy or infant immune function:

Response: We have reviewed and removed Reference #72, replacing it with a more appropriate citation that directly relates to neonatal immune function and allergy outcomes.

Comment 10: Define HMO (lines 356–359):

Response: “HMO” has been defined at its first mention as “Human Milk Oligosaccharides”.

Reviewer 2 Report

Comments and Suggestions for Authors

Maternal microbiome
As a preliminary observation I strongly recommend to clarify that this review is analysing the bactreia content and not the bacterial genomic content. The use of the word “microbiome” instead of “microbiota” can be confusing.
I am quite puzzled by the exclusion of papers due to the unavailability of the full text. How many? This exclusion criterion might reduce the significance of the study.
Section 3.1 Geographical variation in maternal microbiome influence and its impact on neonatal 201 immunity is not rally dealing with geographical differences but with different procedures; i.e probiotic supplementation during pregnancy may occur all over the world, not in Finland only, etc Please rephrase the title of the section
Authors do not take into account several “confounding factors” that can interact with the mode of delivery, i.e. breast feeding  vs formula feeding has to also supported by observations on HMO content
(see Inchingolo et al, 2024, https://doi.org/10.3390/ijms25021055)
Minor points
I suggest a revision of the text for the English usage

Comments on the Quality of English Language

The grammar is good but the structure of some sentencs must be improved

Author Response

Response to Reviewer #2

Dear respected Reviewer#2,

We sincerely appreciate your detailed review and constructive feedback on our manuscript.

Please find below the point-by-point response to your valuable comments, noting that the changes in the manuscript made in response to your comments are highlighted in Green:

Comment 1: As a preliminary observation I strongly recommend to clarify that this review is analysing the bactreia content and not the bacterial genomic content. The use of the word “microbiome” instead of “microbiota” can be confusing:

Response: Our goal was to balance scientific accuracy with readability. However, recognizing the journal’s preference for a more formal academic tone, we have thoroughly revised the manuscript to enhance technical precision and scientific rigor. Additionally, the manuscript has been professionally edited using MDPI Author Services to ensure consistency and adherence to academic standards.

Comment 2: I am quite puzzled by the exclusion of papers due to the unavailability of the full text. How many? This exclusion criterion might reduce the significance of the study:

Response: Thank you for this important observation. As outlined in the revised Methods section, out of the 4,199 articles screened for eligibility, a total of 3,219 studies were excluded. Of these, 412 studies (9.8%) were specifically excluded due to unavailability of the full text, which we now explicitly state in the manuscript. The remaining exclusions were due to duplicate records, research flaws, study populations not including pregnant individuals, a lack of analysis of the maternal microbiota’s effects on neonatal immunity and allergies, non-English language, or insufficient data for assessment. We have also added this specific number to both the text and Figure 1 of the study selection flow diagram to increase transparency.

Comment 3: Section 3.1 Geographical variation in maternal microbiome influence and its impact on neonatal 201 immunity is not rally dealing with geographical differences but with different procedures; i.e probiotic supplementation during pregnancy may occur all over the world, not in Finland only, etc Please rephrase the title of the section:

Response: We appreciate this constructive suggestion. We have rephrased the title of Section 3.1 to more accurately reflect the content.

Comment 4: Authors do not take into account several “confounding factors” that can interact with the mode of delivery, i.e. breast feeding  vs formula feeding has to also supported by observations on HMO content

(see Inchingolo et al, 2024, https://doi.org/10.3390/ijms25021055):

Response: We have revised the relevant section to incorporate discussion of breastfeeding vs. formula feeding as an interacting factor with delivery mode and added specific mention of human milk oligosaccharides (HMOs) as highlighted in the study by Inchingolo et al. (2024).

Comment 5: I suggest a revision of the text for the English usage:

Response: We confirm that the manuscript has been revised for language quality using MDPI Author Services to ensure clarity, grammatical accuracy, and consistency throughout the text.

Round 2

Reviewer 1 Report

Comments and Suggestions for Authors

Statement at line 53 suggests that cesarean section impacts the maternal microbiome. If there is literature showing this, it should be directly cited and not simple referred to with a review/meta-analysis article. By the way, this article does not support the statement that cesarean delivery impacts the maternal microbiota in any way. It is the infant microbiota that is impacted.

Font in Figure 1 is too small.

Paragraph beginning at line 204 refers vaguely to “baby’s immune system” development, but this is highly imprecise. The authors need to be much more rigorous throughout this article with regard to what allergy is in terms of immune function, and be very specific about how immune function was assessed in the articles that they use. Other examples of this or similar problem with language imprecision: lines 259, 278, 281, 289, etc

Relatedly, the authors refer throughout the article to “promotion” of infant immune development, but I believe what they really mean is influence of microbiota on avoidance of hypersensitivity to antigens, which is the definition of allergy. It deserves mention that the term allergy is never defined in the article.  This must be included.

Author Response

Dear respected Reviewer#1, we sincerely appreciate your detailed review and constructive feedback on our manuscript.

Kindly, find the below point-by-point response to your valuable comments noting that the changes in the manuscript in response to your comments were highlighted in Red”:

Comment 1: Statement at line 53 suggests that cesarean section impacts the maternal microbiome. If there is literature showing this, it should be directly cited and not simple referred to with a review/meta-analysis article. By the way, this article does not support the statement that cesarean delivery impacts the maternal microbiota in any way. It is the infant microbiota that is impacted.

Response: Upon review, we agree that the original phrasing may have unintentionally implied that cesarean section affects the maternal microbiota. The statement has been corrected to reflect that cesarean delivery impacts the neonatal microbiota, not the maternal one. Additionally, we have removed the inaccurate citation and replaced it with primary literature that directly supports the influence of delivery mode on infant gut microbiota.

Comment 2: Font in Figure 1 is too small.

Response: We have revised Figure 1 and increased the font size for all labels and axis elements to enhance readability in both digital and print formats. The revised figure is now included in the resubmission files.

Comment 3: Paragraph beginning at line 204 refers vaguely to “baby’s immune system” development, but this is highly imprecise. The authors need to be much more rigorous throughout this article with regard to what allergy is in terms of immune function, and be very specific about how immune function was assessed in the articles that they use. Other examples of this or similar problem with language imprecision: lines 259, 278, 281, 289, etc. 

Response: To improve scientific accuracy, we have revised the phrasing throughout the manuscript to avoid vague terms such as “baby’s immune system” and replaced them with more precise descriptions.

Comment 4: Relatedly, the authors refer throughout the article to “promotion” of infant immune development, but I believe what they really mean is influence of microbiota on avoidance of hypersensitivity to antigens, which is the definition of allergy. It deserves mention that the term allergy is never defined in the article.  This must be included. 

Response: We agree that the term “promotion of immune development” may have been used too broadly in the context of allergy risk and immune tolerance. We have revised such phrasing to more accurately reflect the underlying immunological processes.

We hope that this revision addressed all the respected editor’s and reviewers' concerns and improves the quality of the manuscript. Please let us know if any further clarifications are required.

Thank you for your time and consideration.

Sincerely,

Dr. Arwa Alsharif

25-May-2025
